# Immunotherapy and Cannabis: A Harmful Drug Interaction or Reefer Madness?

**DOI:** 10.3390/cancers16071245

**Published:** 2024-03-22

**Authors:** Brian J. Piper, Maria Tian, Pragosh Saini, Ahmad Higazy, Jason Graham, Christian J. Carbe, Michael Bordonaro

**Affiliations:** 1Department of Medical Education, Geisinger Commonwealth School of Medicine, Scranton, PA 18509, USA; mtian1@geisinger.edu (M.T.); psaini@som.geisinger.edu (P.S.); cjcarbe@geisinger.edu (C.J.C.); mbordonaro1@geisinger.edu (M.B.); 2Center for Pharmacy Innovation & Outcomes, Geisinger, Danville, PA 17821, USA; 3Department of Mathematics, University of Scranton, Scranton, PA 18510, USA

**Keywords:** ipilimumab, marijuana, nivolumab, pembrolizumab

## Abstract

**Simple Summary:**

Two Israeli studies about medical marijuana potentially interfering with immunotherapies like nivolumab for cancer treatment have received substantial attention. However, there have been anonymous but detailed concerns about these reports on PubPeer. This team attempted to verify the data analysis and statistics of these two reports and the published correction. Many findings, including some that could impact the statistical conclusions, could not be verified. Of 22 statistical in the prospective report, 4 could not be repeated using the same statistics or with the provided N. The *p*-value on 17 corresponded with that of a different statistical test than was listed in the methods. Re-analysis also identified some previously unreported significant differences (e.g., age) between cannabis users and non-users at baseline. Further study of the safety of immunotherapy and cannabis combination may be warranted using patient groups that have been matched on key demographic and medical variables.

**Abstract:**

A retrospective (N = 140) and a prospective (N = 102) observational Israeli study by Bar-Sela and colleagues about cannabis potentially adversely impacting the response to immunotherapy have together been cited 202 times, including by clinical practice guidelines. There have also been concerns on PubPeer outlining irregularities and unverifiable information in their statistics and numerous errors in calculating percentages. This reanalysis attempted to verify the data analysis while including non-parametric statistics. The corrected prospective report contained 22 *p*-values, but only one (4.5%) could be verified despite the authors being transparent about the N and statistics employed. Cannabis users were significantly (*p* < 0.0025) younger than non-users, but this was not reported in the retrospective report. There were also errors in percentage calculations (e.g., 13/34 reported as 22.0% instead of 38.2%). Overall, these observational investigations, and especially the prospective, appear to contain gross inaccuracies which could impact the statistical decisions (i.e., significant findings reported as non-significant or vice-versa). Although it is mechanistically plausible that cannabis could have immunosuppressive effects which inhibit the response to immunotherapy, these two reports should be viewed cautiously. Larger prospective studies of this purported drug interaction that account for potential confounds (e.g., greater nicotine smoking among cannabis users) may be warranted.

## 1. Introduction

The National Academy of Sciences (NAS) 2017 cannabis report [1] provides key information for oncology patients who would like their decision of whether to use medical marijuana to be empirically informed. There has been substantial or conclusive evidence that oral cannabinoids are effective for chemotherapy-induced nausea and vomiting. Similarly, the NAS rated the strength of the evidence of cannabis or cannabinoids being effective for the treatment of chronic pain as substantial/conclusive. However, the evidence was rated as limited that cannabis was effective for improving anxiety symptoms. There was moderate evidence that cannabis and cannabinoids caused a small increased risk for the development of depressive disorders [1]. The evidence was rated insufficient to support or refute cannabinoids as an effective treatment for cancer-associated anorexia–cachexia syndrome [1]. There was also limited evidence of a statistical association between cannabis smoking and decreased production of inflammatory cytokines in healthy individuals [1]. The American Cancer Society does not take a position for or against the use of medical cannabis [2].

Cancer is a complex disease process traditionally managed with medical treatments that have progressed from surgical resection, radiation therapy, chemotherapy, and targeted drug therapy to, most recently, immunotherapy [3,4]. Unlike traditional therapy, immunotherapy utilizes monoclonal antibodies, small molecule drugs, adoptive cell therapy, oncolytic viruses, and cancer vaccines to activate the body’s innate and adaptive immune responses to produce anticancer effects in order to kill and eliminate tumor cells [4]. However, tumor cells have adapted the ability to express inhibitory “checkpoint” proteins (i.e., cell surface signaling molecules normally expressed in healthy cells that safeguard against dysregulated immunity capable of subjecting the host to immunodeficiency, autoimmunity, malignancy, or harmful immune responses to infectious agents), resulting in decreased function of antigen-specific T cells and ultimately preventing T cells from recognizing and attacking cancer cells [3,5]. Checkpoint inhibition therapy is a form of cancer immunotherapy that employs antibodies against T cell- or antigen-presenting cell surface regulators of immune cell inhibition (i.e., checkpoint inhibitors), then activates cytotoxic T cells to assist in the killing of tumor cells [5]. Two major immune checkpoint pathways that are currently targeted in oncologic immunotherapeutics are the cytotoxic T-lymphocyte-associated antigen 4 (CTLA-4), which regulates T-cell proliferation primarily in lymph nodes early in the immune response, and the programmed cell death protein (PD-1) pathways, which suppress T cells in peripheral tissues after the immune response [5,6]. Commonly employed immunotherapies include ipilimumab, nivolumab, pembrolizumab, atezolizumab, avelumab, and durvalumab [7].

Two observational studies conducted in Israel and published in 2019 [8] and 2020 [9] identified a potential pharmacodynamic drug interaction between immunotherapy and cannabis. The first, a retrospective report, had a moderate sized sample and compared 89 patients that received the anti-programmed death-1 (PD-1) monoclonal antibody immunotherapy nivolumab to 51 that received nivolumab and cannabis [8]. Multivariate analyses determined that cannabis was associated with a decreased response rate to nivolumab, with an odds ratio of 3.1. A Google Scholar search conducted on 10 March 2024 found that this paper [8] has received 124 citations, including by two clinical practice guidelines [10,11] (Appendix A). However, despite the results section noting that “As shown in Table 2, no significant difference was found between the two groups in aspects of demographic and medical characteristics.” [8], an anonymous PubPeer posting in December of 2023 re-analyzed Table 2 [8] using the same statistic and claimed to find four significant unreported baseline group differences, one trend (*p* = 0.06) towards a baseline group difference in smoking (a well-established risk factor for a variety of cancers) [12], five mistakes calculating percentages, and four rounding oversights [13]. A subsequent prospective observational report compared 68 immunotherapy patients with 34 immunotherapy patients taking cannabis. The median overall survival was 6.4 months among cannabis users versus 28.5 months for cannabis non-users (*p* < 0.001) [9]. A Google Scholar search completed on 20 March 2024 determined that this study [9] has received 78 citations (Appendix A). However, there have again been detailed anonymous PubPeer postings [14,15] regarding both the original report and the published correction [9]. These claim many difficulties repeating the non-parametric analyses, three errors in determining percentages, and nineteen rounding issues [14,15]. The purported challenges in transparently calculating and interpreting the bi-variate analysis could challenge the conclusion that “no statistically significant differences were found in the baseline demographic and clinical variables between the study groups” [9].

Two methodological issues in this area may warrant particular attention. First, observational studies with self-selected participants may have important confounds. Many studies have reported that marijuana users have an increased risk for prior nicotine use [16]. Smoking causes not just lung cancer, but also cancers in many other tissues, including the oropharynx, larynx, esophagus, trachea, bronchus, stomach, liver, pancreas, kidney, ureter, cervix, bladder, colorectal and acute myeloid leukemia [12]. The prospective study [9] did not contain any information on smoking tobacco or whether the groups were similar in terms of this variable. It would be unfortunate to conclude that marijuana has negative health outcomes if these were driven by the well-established consequences of tobacco smoking [12]. Second, the analysis of a 2 × 2 table for non-parametric variables is often completed with a chi-square test [17], although other statistics have been developed for when there is a small N. Different practices have evolved since Karl Pearson developed the chi-square in 1900, including using Fisher’s exact test (henceforth Fisher’s, developed in 1930) or chi-square with Yates correction for continuity (henceforth Yates, developed in 1934, when the smallest cell had an observed value < 5 [18]. Yates has been criticized for over-correcting and providing a *p* value that is too large (i.e., overly “conservative”) [19]. The *British Medical Journal* (BMJ) offers more precise guidance [20]: “In fourfold tables (i.e., a 2 × 2 contingency table), a χ^2^ test is inappropriate if the total of the table is less than 20, or if the total lies between 20 and 40 and the smallest expected (not observed) value is less than 5…. An alternative to the χ^2^ test is known as Fisher’s” (Appendix A). As immunotherapy is a first-line or co-first-line treatment for advanced non-small-cell lung cancer [21], metastatic colorectal cancer [22], advanced cutaneous melanoma [23], and advanced kidney cancer [24]), and as medical cannabis is often used by cancer patients [24], the goal of this report was to reanalyze the data from these two reports [8,9], including the correction [9].

## 2. Materials and Methods

This investigation involved an effort to verify the statistical analysis based on the information contained in the retrospective report [8], the prospective report [9], and the corresponding correction of Table 1 [9]. Most of the re-analyses were completed on the 2 × 2 nonparametric tests and determined whether using the reported N and same test would result in the same *p*-value. If the reported *p*-value could be reproduced using the same statistic as that in the original [8,9] methods, this was interpreted as verified. If the reported *p*-value could only be reproduced using a different statistic than that listed in the methods, this was interpreted as misreported. If the *p*-value could not be reproduced with at least three different statistics (chi-square, Fisher’s, or Yates), this was interpreted as unverified. Information about the smallest expected N and the total N was obtained to apply the BMJ guidance [20] on which the non-parametric statistic was recommended. Additional analyses were also completed regarding whether the percentages were accurately reported using Microsoft Excel. Nonparametric analyses were conducted with GraphPad Prism (version 10.2, Boston, MA, USA) [25], with the smallest cell expected values determined in [26]. Pilot testing determined that 2 × 2 nonparametric analyses produced identical *p*-values for Prism, SPSS (version 21, Chicago, IL, USA), and SAS (version 8.3, Cary, NC, USA). A between-groups *t*-test was completed with the mean age, SD, and N provided [8] using GraphPad Prism [27].

## 3. Results

The original report provided sufficient information that an attempt could be made to verify the previously published Tables 1–4 [8], Tables 1–3 [9], and the corrected Table 1 [9].

### 3.1. Taha et al. The Oncologist 2019;24:549–554 [8]

The statistical analysis section noted that “Chi-square test was used to determine the difference between patients’ characteristics in both groups” and “Two-tailed *p* values of 0.05 were considered statistically significant” [8]. The original Table 1 [8] for THC ≥ 10 (yes or no) by progressive disease (yes or no) reported a chi-square *p* = 0.393. Reanalysis revealed a chi-square *p* = 0.2165. However, Fisher’s *p* = 0.3932. Similarly, CBD ≥ 1 (yes or no) by progressive disease reported a chi-square *p* = 0.116. Reanalysis showed a chi-square = 0.0885 but a Fisher’s *p* = 0.1161, indicating that both *p*-values on Table 1 [8] were misreported.

Although not reported as significant in Table 2 [8], the members of the immunotherapy + cannabis group were 5.7 years (significantly) younger than those in the immunotherapy-only group (*t*(138) = 3.137, *p* = 0.0021, Figure 1A). Smoking was 16.5% more frequent in the immunotherapy + cannabis (56.9%) group than the immunotherapy-only group (40.4%), but this difference was not quite significant (χ^2^(1) = 3.512, *p* = 0.0609, Figure 1B).

### 3.2. Bar-Sela et al. Cancers 2020:12:2447 [9]

The statistical analysis section noted “A series of χ^2^ tests or Fisher’s exact tests were conducted to analyze the differences between patients’ characteristics in both groups.… We computed 2-tailed *p*-values, where *p* < 0.05 was considered a statistically significant result” [9]. Table 1 [9] reported a non-significant *p*-value of 0.05178 for the non-parametric analysis (chi-square or Fisher’s) of whether immunotherapy was received as the first-line vs. second+-line treatment. However, recalculation determined that the chi-square *p* = 0.0307 (Figure 1C) and Fisher’s *p* = 0.0334. The cannabis users (76.4%) were significantly more likely at baseline to receive immunotherapy as a second+-line or later treatment than the cannabis non-users (54.4%). As the total N exceeded 40 and the minimum expected cell value was 13, the BMJ guidance [20] indicates that chi-square was the appropriate analysis (Table 1). This result was classified as misreported as the recalculated Yates *p* = 0.0518. Further information on Table 1 may be found in Section 3.3.

The results section [9] noted “liver metastasis of the immunotherapy group (I-G) (I-G 19%) vs. the immunotherapy-cannabis group (IC-G) (67%, *p* = 0.89)”, which, based on this *p*-value, would be interpreted as non-significant. However, reanalysis revealed that this baseline difference was significant (*p* < 0.0001 for chi-square, Fisher’s, and Yates, Figure 1D).

Table 2 [9] reported two trends in *p*-values of abnormal laboratory tests, but both were unverified. For lymphocytes, this was reported as *p* = 0.08, but the calculated two-tailed chi-square was *p* = 0.1199, Fisher’s *p* = 0.1412, and Yates *p* = 0.1793. As the results section [9] mentioned a one-tailed *p*-value, the calculations of one-tailed values were as follows: chi-square *p* = 0.0600, Fisher’s *p* = 0.0890, and Yates *p* = 0.0896. Similarly, for alkaline phosphatase, the reported *p*-value (*p* = 0.09) could not be verified despite six attempts (two-tailed: chi-square = 0.1374, Fisher’s = 0.1468, Yates = 0.2157; one-tailed chi-square = 0.0687, Fisher’s = 0.1089, Yates = 0.1079). Further, of the twelve reported percentages, two (16.7%) contained minor rounding issues (12/68 reported as 17 but calculated as 17.6%, which would round to 18, and 23/34 reported as 67 but calculated as 67.6%, which should round to 68%).

### 3.3. Correction to Bar-Sela et al. Cancers 2020:12:2447 [9]

The correction published in April of 2022 to [9] contained a new Table 1. Of the 22 *p*-values reported from 2 × 2 analyses, 4 (18.2%) could not be replicated with the chi-square or Fisher’s exact test (i.e., the non-parametric statistics listed in the methods) [9], nor with chi-square with Yates correction. However, the *p*-values on 17 statistical tests did correspond to four decimal places with that of a different statistic (Yates) than was listed in the methods. Finally, for renal cell carcinoma, the percentages (+/total) were equal in both groups (5.9%), and the *p*-value (1.000) was the same for the chi-square, Yates, and Fisher’s tests (Table 1). Overall, 4.5% of the *p*-values from Table 1 [9] were verified. Table 1 also shows that four analyses had minimum expected cell values < 5. However, as the total N (102) was well (>2.5 fold) above 40, the BMJ guidance [20] indicates that chi-square would be the appropriate analysis.

Table 1 [9] did not list a *p*-value for the age comparison. However, the data were reported in a format (median, min–max) that precluded reanalysis of whether the cannabis users being three years younger was a significant difference. There were three errors in calculating percentages among cannabis users. Chronic diseases = 0 (13/34) was reported as 22.0%, but calculated as 38.2%. High blood pressure (13/34) was reported as 34.1%, but calculated as 38.2%. Brain metastasis (8/34) was reported as 13.2%, but calculated as 23.5% (Appendix A). The corrected table [9] contained 19 rounding errors, mostly due to floor rounding of 0.1% (e.g., 55/68 reported as 80.8% but calculated as 80.9%, Appendix A).

## 4. Discussion

The observational studies [8,9] identifying a potential drug interaction between cannabis and immunotherapy have been widely cited, including in clinical practice guidelines [10,11] and in a publication geared to the general public [2] (Appendix A). This report determined that an appreciable subset of the statistics contained in [8,9] could not be verified, as was generally consistent with the PubPeer reports [13,14,15]. There are a few possibilities that might be able to account for only 4.5% of the *p*-values in the correction [9] being verified and the many inconsistencies. Analyses in the retrospective report were completed with SPSS, version 21 [8]. The prospective study listed both SAS and R (https://www.r-project.org/ accessed 18 March 2024) [9]. Although we used GraphPad for our analyses [25,27], we feel that it is unlikely that different software would produce such disparate results. Another possibility is that research with patients is challenging, and there was a small amount of missing data. We feel this explanation is also unlikely because the correction listed the N for each cell for four variables [9], and there were no missing data. Similarly, verification of the percentages (Appendix A) did not indicate that the denominator was decreased. However, we cannot discount this possibility for other variables or tables. Another possibility, which we feel is more likely because the *p*-values corresponded to four decimal places on fifteen occasions (Table 1), is that the methods section listed one statistic (chi-square), but a slightly different statistic (chi-square with Yates) was completed. The PubPeer response by a middle author also indicated that Yates was completed [14]. Notably, the BMJ guidance [20] attempted to clarify when to use chi-square, Fisher’s, and Yates. Table 1 suggests that Yates may have been uniformly used, independent of the minimum expected value or the total N. Although the methods section listed “We computed 2-tailed *p*-values”, as the attempted verification produced *p*-values that were approximately twice as large (and the results section alluded to a one-tailed test), it is also possible that the reported *p*-values were one-tailed in the previously published Table 2 [9]. Similarly, while the methods section [8] states that “Chi-square test was used to determine the difference between patients’ characteristics in both groups.”, it is likely that some of the reported *p*-values were from Fisher’s. It is important for the readers to be informed of which nonparametric statistic was completed so that they can assess the Type I error rate. There is a general consensus that Yates is overly cautious in its desire to avoid a type I error [19]. Fisher’s should not be reported if the total N of all four cells is above 40 (the N was 140 in [8]) and the expected N was ≥5 [21]. Although choosing which non-parametric statistic to use can be challenging when different resources have contradictory recommendations [19,20] (Appendix A), at the very least, we believe that telling the audience in the methods that you will run statistic A but then reporting in the results the findings from statistic B is a non-trivial oversight.

In evaluating the prior findings of a possible cannabis–immunotherapy interaction [8,9], it is important to be cognizant that there were higher rates of lung, laryngeal, esophageal, pancreatic, urinary, stomach, hepatic, colorectal, renal, bone marrow, and cervical cancers in patients who smoked [12]. Smoking tobacco introduces the human body to a complex interplay between thousands of chemical constituents, including dozens of known carcinogens [28,29]. The dangers of this chemical cocktail are well documented, as 40% of new cancer diagnoses are attributed to cigarette smoking [30]. Furthermore, smoking not only increases the relative risk of developing certain cancers, but it has also been associated with poor treatment outcomes [31]. A retrospective review of 439 patients with non-small cell lung cancer found heavy smoking to be associated with poorer overall survival (hazard ratio (HR) = 1.4). This relationship existed regardless of cancer staging [32]. Reported relationships between smoking and melanoma are inconsistent, with some reports demonstrating a paradoxical decreased risk of melanoma in smokers [33,34,35]. Despite this, a cohort study of 6279 patients with stage 1 or stage 2 melanoma reported a greater risk of melanoma-associated death in current smokers (HR = 1.5) [36]. A meta-analysis including 1326 cases of renal cell cancer (RCC) from cohort studies and 6032 cases of RCC from case–control studies reported a relative risk of 1.4 for the development of RCC in ever-smokers [37]. Furthermore, a systematic review and meta-analysis including 343,993 cases of RCC reported that current smoking was associated with increased disease-specific mortality (HR = 1.5) and worsened progression-free survival (HR = 2.9) [38].

Although there were many differences between our findings are those reported earlier [8,9], we are not suggesting that anyone has engaged in anything nefarious. We commend the earlier reports [8,9] for addressing a timely and important topic. However, it may also be valuable to prevent situations like this in the future. If the authors are going to engage in an atypical practice (e.g., floor rounding [14]), it would be beneficial to briefly document this in the methods section. It is also possible that floor rounding was employed for reporting the *p*-values. About one-third of leading biomedical journals have reported rarely or never using specialized statistical reviews [39]. As the statistical analysis section contained “A series of χ2 tests or Fisher’s exact tests (when the assumptions of the parametric χ2 test (sic) were not met)” [9], if there was a statistical reviewer, this person may not have been reading carefully. Emails to the editors asking whether these manuscripts [8,9] received a statistical review did not receive an affirmative response. Further, it would be well beyond standard practice to expect a reviewer (often a volunteer) to re-run all the analyses. In psychology, it is common to list the test statistic, the degrees of freedom, and the *p*-value. An analysis of a quarter million psychology papers revealed that half contained inconsistencies where the reported test statistic and *p*-value did not correspond. Further, one-seventh of papers included gross inconsistences defined as “the reported *p*-value was significant and the computed *p*-value was not, or vice versa” [40]. Some experimental psychology journals use software to identify statistical irregularities [41]. We are not aware of any biomedical journals which are employing similar software to assess percentage calculations, including rounding errors or non-parametric analyses.

Age was not reported as statistically significant in [8], but had a significant difference (*p* < 0.002) in our analysis (Figure 1A). This age difference could reflect that cannabis is psychoactive and providers in Israel may be less likely to prescribe cannabis to older patients [42]. Alternatively, cannabis users could have had a more aggressive or more advanced form of cancer at baseline. As the cannabis users were much more likely to have liver metastasis at baseline (Figure 1D), cannabis may simply be a proxy for a high-burden symptomatic disease [41]. Whether immunotherapy was received as a first- or second-line treatment was reported as non-significant (*p* > 0.050) in Table 1 [8], and again in [9]. However, re-analysis revealed that this baseline difference was significant (Figure 1C). Both of these findings could be interpreted as gross inaccuracies. In one high-profile instance of a manuscript with multiple issues, there was a twelve-year interval between the original MMR vaccine and autism study [43] and the retraction [44]. Journals, including those with reasonable open-access fees (e.g., *Cancers* is currently CHF 2900 [45]; *The Oncologist* is currently USD 3669 [46]), need to have sufficient staff to deal with these concerns in a timely fashion. A practice that might be helpful moving forward is to include the full data and a detailed data dictionary as Appendix A, either with the published manuscript or with a preprint, to allow a second party to verify all analyses. The author instructions of *Cancers* include “We encourage all authors of articles published in MDPI journals to share their research data.” [46]. Similarly, the author guidelines for the retrospective journal [47] notes that “*The Oncologist* strongly encourages authors to make all data and software code on which the conclusions of the paper rely available to readers.” An email to the corresponding author requesting the original data was not responded to. It would also be informative to list the N for all cells in each analysis in a Appendix A to facilitate statistical verification.

Basic science investigations are crucial, as they can avoid the many potential confounds (e.g., age, disease severity, smoking [12], Figure 1) that are challenging to overcome with observational reports. A recent report attempted to replicate and extend the studies [8,9] in two ways. First, tetrahydrocannabinol did not impact the enhanced survivability of anti-programmed death ligand 1 antibody treatments in a murine colorectal model. Second, although the advanced non-small-cell lung cancer patients using cannabis (N = 102) were significantly younger, more likely to have brain metastasis, and marginally more likely to have liver metastasis (*p* = 0.06) at baseline than those that did not (N = 99), cannabis did not significantly impact the survivability following pembrolizumab [42].

In balancing the benefits and harms of cannabis for cancer patients [1,2,42], a broader perspective is useful [48,49,50,51,52,53,54,55]. There has been appreciable work focused on the anti-cancer activity of cannabinoids, including possible use in potentiating immunotherapy [48,49,50,51,52]. Cannabinoids, including cannabigerol (CBG), cannabidiol (CBD), and tetrahydrocannabinol, have been shown, in vitro and in vivo, to inhibit cancer cell proliferation and metastasis, while also promoting apoptosis and suppressing cancer-related angiogenesis [49,50,51]. Cannabinoids may also play a positive role in regulating aberrant cellular metabolism, which is characteristic of cancer [51]. With respect to the role of cannabinoids in potentially potentiating immunotherapy in colorectal cancer, these agents may boost immunogenicity through cytotoxic effects on cancer cells and the subsequent release of antigens, as well as reprograming immune cells to target tumors [51]. Given the various mechanisms by which cannabinoids may have anti-cancer effects, it is important not to dismiss potential benefits based on possibly unverifiable statistical interpretations of data. In addition, the Taha [8] and Bar-Sela [9] studies involved a limited number of cancer types, which should not be considered universally applicable for cancer immunotherapy in general. It should also be noted that effects of cannabinoids on immunotherapy may not be influenced only by cancer type, but also by the concentration of the relevant agents. For example, while higher concentrations of THC had immunosuppressive effects in vitro and in vivo, lower concentrations were immunostimulatory [53]. All of these facts recommend caution in the interpretation of the Taha [8] and Bar-Sela [9] studies, as well as other naturalistic investigations where the cannabis route of administration and THC concentration are not homogenous.

The ability of cannabinoids to influence carcinogenesis and response to immunotherapy, particularly with respect to the colon, could be modulated by nutritional factors, synthetic organic agents, and the gut microbiota. For example, bioactive plant compounds such as flavonoids affect colon tumorigenesis and anti-cancer therapeutics [56]. Xanthohumol demonstrated greater anti-tumor effects on colorectal cancer (CRC) cell lines than those of the chemotherapeutic agent 5-fluorouracil (5-FU); while apigenin and luteolin demonstrated lesser anti-tumor effects, these flavonoids exhibited positive synergism, boosting the efficacy of 5-FU [56]. Since these agents are also known to reduce side effects of CRC chemotherapy, they have a double utility, both promoting anti-tumor activity and making chemotherapy more bearable for the patient. More generally, bioactive food compounds, including those associated with the Mediterranean diet, can affect cancer-related microRNAs (miRNAs) to suppress tumorigenesis, while high-fat diets have an opposing, pro-tumorigenic effect [57]. Given the possible anti-tumor effects of cannabinoids and their use to reduce the side effects of cancer treatment, positive interactions between cannabinoids and nutritional supplementation need to be explored, including with respect to cancer immunotherapy. Other agents may be important adjuncts to CRC therapy in conjunction with cannabinoids. For example, upregulation of the ion channel protein TRPM8 is associated with a poor outcome in CRC patients, and the TRPM8 ligand WS12 represses colon tumorigenesis in mice, at least in part through downregulation of canonical Wnt signaling [58]. Thus, synthetic organics like WS12 may synergize with cannabinoids in the treatment of CRC.

The gut microbiota can influence the development of gastrointestinal cancer, and differences in the microbiota are associated with the up- or downregulation of a number of non-coding RNAs, including miRNAs, linked to the development of gastric or colon cancer [59]. As bioactive food compounds can also alter cancer-related miRNAs [57], and diet can alter the microbiota itself, it is likely that diet and the microbiota synergize to influence CRC risk. Further, interactions between the gut microbiota, dietary fiber, and colon tumorigenesis have been identified [60]. Thus, diet and nutrition synergize with the underlying microbiota to affect tumorigenesis, and these effects could influence cannabinoids’ effects on tumorigenesis and on treatment (e.g., immunotherapy), as well as on the adverse effects of anti-cancer treatments (e.g., nausea). Therefore, when considering the question of cannabinoids and cancer, the context of the patient’s microbiota, as well as nutrition, are variables that must be taken into careful consideration. An incomplete understanding of these factors, or a failure to account for these variables, can contribute to misleading conclusions with respect to the negative effects of cannabinoids on immunotherapy and other cancer treatments.

One limitation of this report is that only a subset of the originally reported findings could be assessed for veracity because the full raw data were unavailable. Therefore, we cannot infer whether the figures contained verifiable, misreported, or unverifiable information. As noted above, we cannot discount the possibility that the Israeli team [14] also used floor rounding to report their *p*-values (e.g., *p* = 0.0896 reported as 0.08), so some analyses that were classified as unverified may in fact have been misreported instead. There are many statisticians who would argue that it is ridiculous to interpret the results of a study with *p* = 0.055 differently than one with *p* = 0.045. Dividing the results of hypothesis tests into “significant” and “non-significant” is both unhelpful and outdated [52]. Although we would concur with the thrust of this argument, findings like Table 1 [8], where age is described as non-significant (i.e., *p* > 0.05) but then has a recalculated *p*-value of 0.0021, or the liver metastasis, which was reported as “*p* = 0.89” but calculated as *p* ≤ 0.0001 [9], are more concerning, as they obscure the reader’s ability to accurately determine whether the cannabis user and non-user groups were equivalent at baseline.

## 5. Conclusions

In conclusion, reanalysis of a subset of the reported information in the very influential observational reports purporting to show a harm-inducing drug interaction between immunotherapy and cannabis [8,9] was unable to verify many of the analyses, and there were some gross inaccuracies. This reanalysis and Figure 1 call into question the conclusion that “no statistically significant differences were found in the baseline demographic and clinical variables between the study groups” [9]. Future prospective studies on this topic could consider matching key demographic or medical condition variables. Due to the well-known contribution of smoking to cancer in a variety of tissues [12], excluding participants who are current or former tobacco smokers, or at least stratifying a homogenous sample with one cancer type accordingly, might be informative. Perhaps it would be more important to strengthen practices [54] in order to prevent similar situations [55] from occurring. This could include registering the methods, making the raw data publicly available, and developing and implementing statistical software [41] that could assist manuscript reviewers in identifying irregularities in the reporting of descriptive and bivariate analyses.

## Figures and Tables

**Figure 1 cancers-16-01245-f001:**
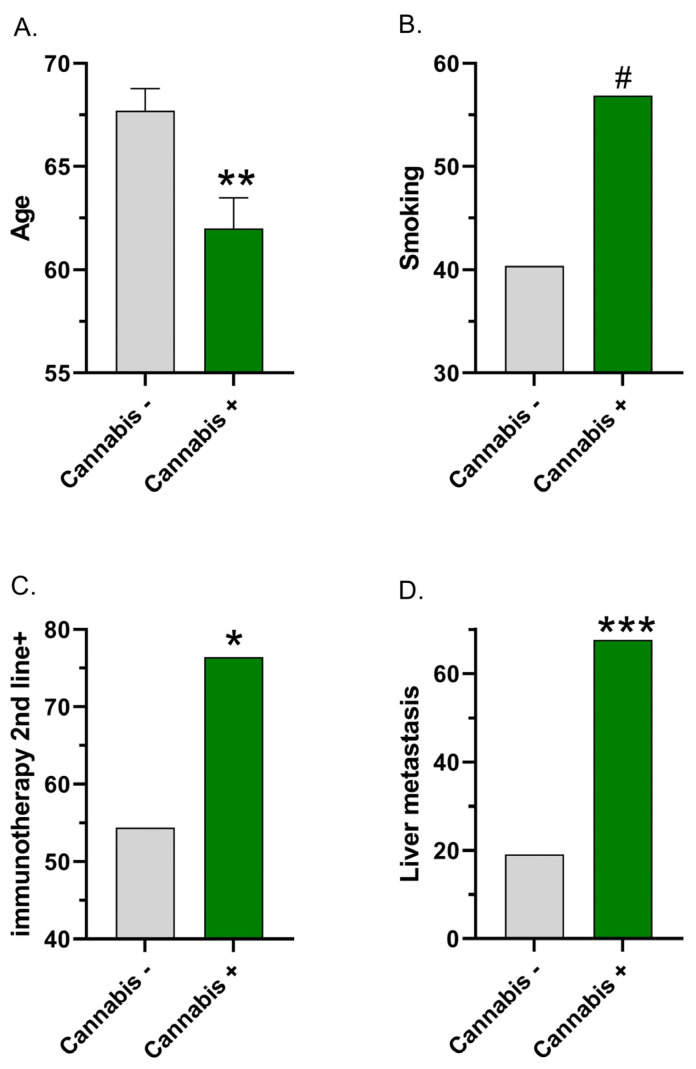
Baseline group differences between cannabis non-users (−) and users (+) based on re-analysis of the reported data in [8] (**A**,**B**) Age (**A**); smoking (**B**); and [9] (**C**,**D**) whether immunotherapy was received as a second-line treatment; (**C**) liver metastasis (**D**). ^#^ chi-square *p* ≤ 0.061, * chi-square *p* < 0.05, ** *t*-test *p* < 0.005, *** chi-square *p* < 0.0001.

**Table 1 cancers-16-01245-t001:** Results of re-analysis of the Bar-Sela correction [9], with total N = 102, evaluating potential adverse effects of cannabis on immunotherapy using ^C^ chi-square, ^F^ Fisher’s, or ^Y^ Yates non-parametric statistics. Minimum expected cell N determined in [26]. The British Medical Journal (BMJ) guidance on the recommend non-parametric statistic is reported [20]. atezo: atezolizumab, COPD: chronic obstructive pulmonary disease, ECOG: Eastern Cooperative Oncology Group, LC: lung cancer, nivo: nivolumab.

Variable	Smallest Cell N Observed (Expected)	BMJ Statistic	Reported p	Calculated p	Interpretation
Gender	10 (10.67)	chi-square	0.9399 ^C or F^	0.9399 ^Y^	Misreported
ECOG	10 (7.67)	chi-square	0.3568 ^C or F^	0.3568 ^Y^	Misreported
Chronic diseases = 0 Chronic diseases = 1 Chronic diseases = 2+	13 (11.67) 7 (7.67) 14 (14.67)	chi-square chi-square chi-square	0.7124 ^C or F^ 0.9332 ^C or F^ 0.9437 ^C or F^	0.7124 ^Y^ 0.9332 ^Y^ 0.9437 ^Y^	Misreported Misreported Misreported
Chronic heart disease	5 (7.67)	chi-square	0.2762 ^C or F^	0.2762 ^Y^	Misreported
Diabetes	6 (7.67)	chi-square	0.5576 ^C or F^	0.5576 ^Y^	Misreported
High blood pressure	13 (15.67)	chi-square	0.3612 ^C or F^	0.3612 ^Y^	Misreported
COPD	3 (4.00)	chi-square	1 ^C or F^	0.5145 ^C^, 0.7463 ^F^, 0.7445 ^Y^	Unverified
Hyperlipidemia	7 (10.00)	chi-square	0.2491 ^C or F^	0.2491 ^Y^	Misreported
Other disease	0 (-)	chi-square	1 ^C or F^	0.3125 ^C^, 0.5512 ^F^, 0.8007 ^Y^	Unverified
Non-small-cell LC	14 (15.00)	chi-square	0.8325 ^C or F^	0.8325 ^Y^	Misreported
Melanoma	9 (11.33)	chi-square	0.414 ^C or F^	0.4140 ^Y^	Misreported
Renal cell carcinoma	2 (2.00)	chi-square	1 ^C or F^	1.000 ^C,F,Y^	Verified
Other malignancy	3 (1.67)	chi-square	1 ^C or F^	0.1946 ^C^, 0.3300 ^F^, 0.4175 ^Y^	Unverified
Brain metastasis	8 (6.67)	chi-square	0.6593 ^C or F^	0.6593 ^Y^	Misreported
Lungs metastasis	11 (13.33)	chi-square	0.4303 ^C or F^	0.4303 ^Y^	Misreported
Liver metastasis	11 (8.00)	chi-square	0.2157 ^C or F^	0.2157 ^Y^	Misreported
Immunotherapy 1st line	8 (13.00)	chi-square	0.05178 ^C or F^	0.0518 ^Y^	Misreported
Pembrolizumab or nivo	5 (8.67)	chi-square	0.127 ^C or F^	0.1270 ^Y^	Misreported
Ipilimumab and nivo	4 (6.67)	chi-square	0.2517 ^C or F^	0.2517 ^Y^	Misreported
Durvalumab or atezo	1 (2.00)	chi-square	1 ^C or F^	0.3720 ^C^, 0.6607 ^F^, 0.6554 ^Y^	Unverified

## Data Availability

All of the information used in this manuscript is publicly available [8,9].

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
