# Peer review of "Immunotherapy and Cannabis: A Harmful Drug Interaction or Reefer Madness?"

_cancers, 2024, doi:10.3390/cancers16071245_

Round 1

Reviewer 1 Report

Comments and Suggestions for Authors

In the present article the authors have re-analyzed the data from two previously published clinical studies that investigated the effect of cannabis in the immunotherapy of cancer. The study is interesting and provides a new explanation and a new interpretation of data.

More specifically, the two previous studies that were analyzed were published in 2019 and 2020 and the first was a retrospective report while the other was a prospective study. Both were conducted in Israel and included (a) the administration of monoclonal antibodies (nivolumab) and cannabis vs. the nivolumab and (b) the administration of monoclonal antibodies vs antibodies and cannabis.

This study had received a number of comments regarding its statistical analysis and therefore the conclusions at the PubPeer. Thus, the authors wished to perform a new more detailed statistical analysis. With this new analysis, the authors have found different results compared to the published study. Briefly, some differences in cannabis consumption and immunothepary are illustrated in Figure 1. Interestingly, it appears that cannabis consumption is related to increased liver metastasis which is discussed in the text and potential explanation are provided.

Finally, the potential limitations are provided. This study calls for further research on the effects of cannabis on immunotherapy of cancer.

I recommend the publication of the present article without further reviewing.

Author Response

Reviewer #1

  1. In the present article the authors have re-analyzed the data from two previously published clinical studies ... I recommend the publication of the present article without further reviewing.

No adjustments were necessary or made.

Reviewer 2 Report

Comments and Suggestions for Authors

The manuscript is interesting.

I suggest the authors read these articles and insert them, in this way to make it clear that there are also other medicinal plants/compounds to treat colon cancer

Fernández J, Silván B, Entrialgo-Cadierno R, Villar CJ, Capasso R, Uranga JA,

Lombó F, Abalo R. Antiproliferative and palliative activity of flavonoids in

colorectal cancer. Biomed Pharmacother. 2021 Nov;143:112241.

Pagano E, Romano B, Cicia D, Iannotti FA, Venneri T, Lucariello G, Nanì MF,

Cattaneo F, De Cicco P, D'Armiento M, De Luca M, Lionetti R, Lama S, Stiuso P, Zoppoli P, Falco G, Marchianò S, Fiorucci S, Capasso R, Di Marzo V, Borrelli F, Izzo AA. TRPM8 indicates poor prognosis in colorectal cancer patients and its pharmacological targeting reduces tumour growth in mice by inhibiting Wnt/β-catenin signalling. Br J Pharmacol. 2023 Jan;180(2):235-251.

Şahin TÖ, Yılmaz B, Yeşilyurt N, Cicia D, Szymanowska A, Amero P, Ağagündüz D, Capasso R. Recent insights into the nutritional immunomodulation of cancer- related microRNAs. Phytother Res. 2023 Oct;37(10):4375-4397

Do the authors think that the microbiota may be involved in improving the therapeutic effect?

The authors can read this article

Ağagündüz D, Cocozza E, Cemali Ö, Bayazıt AD, Nanì MF, Cerqua I, Morgillo F, Saygılı SK, Berni Canani R, Amero P and Capasso R (2023), Understanding the role of the gut microbiome in gastrointestinal cancer: A review. Front. Pharmacol. 14:1130562

Author Response

Reviewer #2

2. The manuscript is interesting.

I suggest the authors read these articles and insert them, in this way to make it clear that there are also other medicinal plants/compounds to treat colon cancer

Fernández J, et al. Antiproliferative and palliative activity of flavonoids in colorectal cancer. Biomed Pharmacother. 2021 Nov;143:112241.

Pagano E, et al. TRPM8 indicates poor prognosis in colorectal cancer patients and its pharmacological targeting reduces tumour growth in mice by inhibiting Wnt/β-catenin signalling. Br J Pharmacol. 2023 Jan;180(2):235-251.

Şahin TÖ, Yılmaz B, Yeşilyurt N, Cicia D, Szymanowska A, Amero P, Ağagündüz D, Capasso R. Recent insights into the nutritional immunomodulation of cancer- related microRNAs. Phytother Res. 2023 Oct;37(10):4375-4397

This is a key insight. We have added a meaty paragraph with these (and some other) new citations:

The ability of cannabinoids to influence carcinogenesis and response to immunotherapy, particularly with respect to the colon, could be modulated by nutritional factors, synthetic organic agents, as well as the gut microbiota. For example, bioactive plant compounds such as flavonoids affect colon tumorigenesis and anti-cancer therapeutics [Fernandez 2021]. Xanthohumol demonstrated anti-tumor effects on colorectal cancer (CRC) cell lines greater than that of the chemotherapeutic agent 5-fluorouracil (5-FU); while apigenin and luteolin demonstrated lesser anti-tumor effects, these flavonoids exhibited positive synergism is boosting the efficacy of 5-FU [Fernandez 2021]. Since these agents are also known to reduce side effects of CRC chemotherapy, they have a double utility, both promoting anti-tumor activity as well as making chemotherapy more bearable for the patient. More generally, bioactive food compounds, including those associated with the Mediterranean diet, can affect cancer-related microRNAs (miRNAs) to suppress tumorigenesis, while high fat diets have an opposing, pro-tumorigenic effect [Sahin 2023]. Given possible anti-tumor effects of cannabinoids and their use to reduce side effects of cancer treatment, positive interactions between cannabinoids and nutritional supplementation need to be explored, including with respect to cancer immunotherapy. Other agents may be important adjuncts to CRC therapy in conjunction with cannabinoids.   For example, upregulation of the ion channel protein TRPM8 is associated with poor outcome in CRC patients, and the TRPM8 ligand WS12 represses colon tumorigenesis in mice, at least in part through downregulation of canonical Wnt signaling [Pagano 2023]. Thus, synthetic organics like WS12 may synergize with cannabinoids in the treatment of CRC.

Reviewer 3 Report

Comments and Suggestions for Authors

The authors' goal in this paper was to reanalyze the data from the two reports  including the correction.

Abstract: the main aim should be more clearly introduced.

Figure 1 should be definitely  explained in the legend.

Otherwise the paper is well written.

Author Response

Reviewer #3

  1. Abstract: the main aim should be more clearly introduced.

There were some wording adjustments to better set-up the including: There have also been concerns on PubPeer outlining irregularities and unverifiable information in their statistics and numerous errors calculating percentages.

The objective was rephrased too as: This reanalysis attempted to verify the data-analysis including non-parametric statistics.

  1. Figure 1 should be definitely explained in the legend.

As suggested, expanded the legend: “Figure 1. Baseline group differences between cannabis non-users ( - ) and users ( + ) based on re-analysis of the reported data in [8] (A, B), including age (A) and smoking (B), and [9] (C, D) including whether immunotherapy was received as a second-line treatment (C) and liver metastasis (D). #chi-square p < .061, *chi-square p < .05, **t-test p < .005, ***chi-square p < .0001.”
